# Effects of Different Light Spectra on Final Biomass Production and Nutritional Quality of Two Microgreens

**DOI:** 10.3390/plants10081584

**Published:** 2021-07-31

**Authors:** Stefania Toscano, Valeria Cavallaro, Antonio Ferrante, Daniela Romano, Cristina Patané

**Affiliations:** 1Department of Agriculture, Food and Environment (Di3A), Università degli Studi di Catania, 95123 Catania, Italy; stefania.toscano@unict.it; 2IBE-Istituto di BioEconomia, Consiglio Nazionale delle Ricerche, 95126 Catania, Italy; valeria.cavallaro@cnr.it (V.C.); cristinamaria.patane@cnr.it (C.P.); 3Department of Agricultural and Environmental Sciences, Università degli Studi di Milano, 20133 Milan, Italy; antonio.ferrante@unimi.it

**Keywords:** LED, light spectrum, ascorbic acid, chlorophylls, carotenoids

## Abstract

To improve microgreen yield and nutritional quality, suitable light spectra can be used. Two species—amaranth (*Amaranthus tricolor* L.) and turnip greens (*Brassica rapa* L. subsp. *oleifera* (DC.) Metzg)—were studied. The experiment was performed in a controlled LED environment growth chamber (day/night temperatures of 24 ± 2 °C, 16 h photoperiod, and 50/60% relative humidity). Three emission wavelengths of a light-emitting diode (LED) were adopted for microgreen lighting: (1) white LED (W); (2) blue LED (B), and (3) red LED (R); the photosynthetic photon flux densities were 200 ± 5 µmol for all light spectra. The response to light spectra was often species-specific, and the interaction effects were significant. Morphobiometric parameters were influenced by species, light, and their interaction; at harvest, in both species, the fresh weight was significantly greater under B. In amaranth, Chl *a* was maximized in B, whereas it did not change with light in turnip greens. Sugar content varied with the species but not with the light spectra. Nitrate content of shoots greatly varied with the species; in amaranth, more nitrates were measured in R, while no difference in turnip greens was registered for the light spectrum effect. Polyphenols were maximized under B in both species, while R depressed the polyphenol content in amaranth.

## 1. Introduction

Light is one of the major factors for growth. It represents the main signal perceived by plants, and it has been largely demonstrated that different light qualities, light intensity, and photoperiod have broad regulatory effects on the morphogenesis, physiological metabolism, growth and development, and nutritional quality of plants [1,2,3,4]. Plant morphogenesis and its related aspects are mainly regulated by various photoreceptors which are activated by photons in the blue, red, and far-red regions of the light spectrum [5]. Light-emitting diodes (LEDs) are an emerging source of light in protected and indoor cultivations. They have several advantages over conventional lighting systems (fluorescent light, halide metal, high-pressure solid, and incandescent), e.g., long operating lifetime, relatively lower heat emission, high photosynthetically active radiation efficiency, small size, and control of spectral composition. All these advantages make LED an ideal light source for the artificial regulation of plant growth and an easy disposal without any environmental hazards [6]. Moreover, LEDs offer the advantage to emit specific spectral patterns [7] and regulate the light intensities, in accordance with the needs of the plants, optimizing the production processes and/or the production of secondary metabolites [4,7]. For these reasons, LEDs are attracting increasing attention for indoor facilities, vertical farming, and greenhouse productions, especially with leafy vegetables, such as lettuce and rockets [8,9,10,11,12]. According to the manufacturers’ indications and measured light fluence rates, LED lids would require about 32% less energy than fluorescent tubes, per μmol·m^2^·s^−1^ delivered to the plants [13].

Approximately 90% of red and blue light that falls on plant leaves is absorbed. It is well known that those sections of the spectrum strongly influence plant development and physiology [14]. Blue and red light are absorbed by photosynthetic pigments (chlorophylls) and photomorphogenetic (cryptochromes, phytochromes) receptors [15].

Red light influences leaf expansion in red lettuce [16], as well as increases plant height in tomato [17] and in vitro grown chestnut seedlings [18]. Blue light suppresses hypocotyl elongation and induces biomass production [18]. In combination with red and blue light, green light increases plant and leaf growth, as well as early stem elongation [18,19,20]. 

Microgreens are young, tender greens of edible plants that are harvested at the first true leaf stage. Microgreens are much smaller than regular greens, even “baby” greens. They are harvested when plants are no taller than 5 cm, taking about 1–3 weeks after seeding. Microgreens have emerged on the market and become popular for their nutrient concentrations that are higher than those of their mature leaf counterparts [21,22,23]. The attention toward this category of products is confirmed by the very high number of items published about microgreens. Moreover, microgreens have an eye-catching appearance; they can be grown in small spaces and on indoor farms, thus representing a potentially useful addition to urban diets [24]. 

Microgreens are also frequently used to add color and flavor to meals. They have a double function as food and garnish on plates. Micro versions of basil, coriander, chard, beetroot, and red garnet amaranth were originally used to complement the flavor of dishes and as a garnish. Today, since their popularity has widened, people can even buy pots ready to grow your own.

The levels of nutrients in microgreens vary with the species. Nonetheless, they typically have higher levels of vitamin C, vitamin E, and carotenoids than mature plants [25]. Due to their adaptation to different cultivation environments, they can be cultivated in individual households, as well as on a large scale for commercial purposes [26]. Grown in a greenhouse with supplemental lighting and heating, microgreens can be produced throughout the entire year.

Numerous vegetables and crops can be used for microgreen production. Among these, the following are of considerable importance: turnip green and amaranth. Turnip green (*Brassica rapa* L. subsp. *oleifera*) is a member of the Brassicaceae family. The Brassicaceae microgreen effects on health are tied to their high levels of bioactive compounds such as ascorbic acid, carotenoids, tocopherols, and phenolic compounds in addition to glucosinolates and mineral nutrients [23].

Amaranth (*Amaranthus tricolor* L.) is one of the most preferable greens in terms of texture, flavor, appearance, and overall eating quality [25].

Recent studies highlighted the possibility of regulating seedling growth and increasing the content of important nutritional compounds (as glucosinolates in rocket and sugars, proteins, flavonoids, and vitamin C in lettuce) through appropriate regulation of the light spectrum used [9,10,11,12]. 

In recent years, spectral effects of red/blue/red–blue light have been investigated in microgreen species, belonging to different families, e.g., Brassicacceae, Lamiaceae, Apiaceae, Boraginaceae, and Chenopodiaceae [27,28,29,30,31,32]. However, for new and emerging microgreen species, information on plant secondary metabolites profiles and how these bioactive compounds respond to LED spectral quality is lacking. Instead, there is a need, as it is often a species-specific response, to investigate the mechanism of different light spectra on the phytochemical profiles of some microgreens [32]. 

With this in mind, a study was conducted to evaluate the effects of different LED spectra (white, red and blue), on the final biomass and nutritional traits, in two different microgreen species. The hypothesis of the work was to enhance the microgreen composition modulating the light quality.

## 2. Results

### 2.1. Seedling Height and Biomass

Seedling height was influenced by species, light, and their interaction (Table 1, Figure 1). Seedlings of amaranth were significantly smaller than those of turnip greens, under all lights (*p* ≤ 0.001).

However, light exerted a different effect, depending on species (*S* × *L*, *p* ≤ 0.01). In amaranth, seedlings were almost 3 cm tall in W, whilst their height exceeded 4 cm in B and R. In turnip greens, blue light (B) promoted plant growth, resulting in a final seedling height close to 6 cm; in this species, white light (W) adversely affected plant growth, leading to a final height < 5 cm (Figure 1).

Fresh biomass of the single plant varied with species and light but not with their interaction. According to seedling height, turnip greens produced a fresh biomass more than threefold greater than that of amaranth (*p* ≤ 0.001) under the same experimental conditions, revealing a faster growth. Light also affected the biomass accumulation, and fresh weight of the single plant at harvest was significantly greater under blue light in both species (*L*, *p* ≤ 0.001; *S* × *L*, *p* ≥ 0.05). Fresh biomass produced in W and R did not differ at ANOVA. However, dry biomass was the lowest (<4.3%) under red light, in both species, indicating a greater plant water content under these growing conditions. 

When the height/dry biomass (cm·mg^−1^) ratio was calculated, interesting results on plant morphology were obtained (Figure 2). While no differences among light treatments were observed in turnip for the ratio (<2 cm·mg^−1^), the significantly higher value (7 cm·mg^−1^) calculated for amaranth (*S* × *L*, *p* ≤ 0.001) in R with respect to W and B (3.7 cm·mg^−1^, on average) indicates that the same dry matter was distributed over longer plants under red light, i.e., the hypocotyls were thinner under these experimental conditions, contributing to total plant biomass. 

### 2.2. Chlorophyll (a, b, and Total) and Carotenoids 

The content of Chl *a* exhibited a different pattern in relation to species and light. Overall, greater contents were measured in shoots of amaranth (Table 2, Figure 3a). In this species, Chl *a* was maximized in B, whereas, in turnip greens, Chl *a* content did not differ with light (*S* × *L*, *p* ≤ 0.01). No effect of species and light was observed according to ANOVA on Chl *b* content (*S*, *L*, *S* × *L*, *p* ≥ 0.05) (Table 2, Figure 3b). As a result, *total* Chl content followed the same pattern of Chl *a*, being higher in amaranth. In this species, as for Chl *a*, total Chl peaked under blue light, whereas it did not change with light in turnip greens (*S* × *L*, *p* ≤ 0.05) (Table 2, Figure 3c).

Carotenoids were accumulated in a larger amount in microgreens of amaranth (0.10 mg·g^−1^ FW, against 0.08 mg·g^−1^ in turnip greens, *p* ≤ 0.01). They exhibited opposite trends in the two species, in response to light (*S* × *L*, *p* ≤ 0.01) (Table 2, Figure 3d). In amaranth, blue light (B) promoted the biosynthesis of carotenoids, whose content was almost the 30% higher than that in W and R. Such differences were not evidenced in turnip greens, where the content of carotenoids did not change with the light conditions of growth. The chlorophyll *a*/*b* ratio showed that W treatment provided a lower value while the highest ratio was found in the B light treatment. The relationship between total chlorophyll and carotenoids expressed as a ratio did not show any significant difference (Table 2).

### 2.3. Sugar Content

Sugar content varied with the species, being more than 40% greater in turnip greens (>1.3 mg·g^−1^ FW) than in amaranth (<0.7 mg·g^−1^ FW) (Table 3).

Light did not exert any clear effect on this trait (*L*, *p* ≥ 0.05), while the interactive effect with species was not significant (*S* × *L*, *p* ≥ 0.05).

### 2.4. Nitrate Content

The nitrate content of shoots greatly varied with the species (*S*, *p* ≤ 0.001) (Figure 4). Greater amounts were measured in amaranth, where nitrates peaked under red light (>2000 mg·kg^−1^). Lower contents (1583 to 1946 mg·kg^−1^) were measured in this species in W and B, showing no difference according to ANOVA. Unlike amaranth, very low nitrate contents (<705 mg·kg^−1^) were detected in turnip greens, regardless of the light conditions of growth (*S* × *L*, *p* ≥ 0.05). 

### 2.5. Antioxidants and Antioxidant Activity

The content of the main antioxidants (polyphenols, carotenoids, and ascorbic acid) was measured in this experiment. Significant differences were found for total polyphenols in relation to species, light, and their interaction (Table 4, Figure 5). On average across light conditions, the antioxidant content was slightly but significantly higher in turnip greens (>145 mg GAE·100 g^−1^ FW vs. 124 mg·g^−1^ in amaranth). Polyphenols were maximized under blue light (B, >165 mg·g^−1^ FW) in both species. Red light (R) somehow depressed the biosynthesis of polyphenols, leading to a final content that was overall the lowest in amaranth (<75 mg·g^−1^ FW), but did not differ from that in W (128.2 and 135.9 mg·g^−1^ FW in R and W, respectively) in turnip greens (*S* × *L*, *p* ≤ 0.01).

Ascorbic acid (Asc) is another antioxidant that was detected in the microgreens of the two species. Unlike carotenoids, much greater contents of Asc (up to 1.3 mg·g^−1^ FW) were found in turnip greens (*p* ≤ 0.001) (Table 4, Figure 6). Light strongly affected (*L*, *p* ≤ 0.001) the content of this antioxidant, being significantly higher under blue light (B) in both species. A significant *S* × *L* interaction (*p* ≤ 0.001) was found according to ANOVA, indicating that, unlike amaranth, whose microgreens had the same Asc in W and R, white light (W) significantly reduced the accumulation of this metabolite in shoots of turnip greens, whose final content was <0.3 mg·g^−1^ FW.

Antioxidant activity (AA), expressed as DPPH free-radical scavenging activity, was seemingly correlated to Asc (*r* = 0.82*) more than to other antioxidants (*r* = 0.54^ns^ vs. carotenoids, *r* = 0.41^ns^ vs. TPC). As a result, on average across light conditions, higher AA corresponded to turnip greens (up to 260 mg TE·100 g^−1^ FW) with respect to microgreens of amaranth (AA <74%) (Figure 7). Light also exerted a significant effect on this trait, with AA being higher in microgreens grown under blue light (B). However, a significant *S* × *L* interaction (*p* ≤ 0.001) revealed that, while no differences were observed for AA between W and R in amaranth (46 mg TE·100 g^−1^ FW, on average), red light (R) adversely affected the antioxidant activity in amaranth, which was the 55% and 27% lower than AA in B and W, respectively.

### 2.6. Mineral Composition

Multifactorial ANOVA showed that the mineral contents were significantly affected by species and the LED treatments, as well as by their interaction (Table 5). Most of the mineral elements were different in the two species except for Fe and Ni. Amaranth showed higher concentrations of Mg, K, Cu, Zn, and P, but lower concentrations of Na, Ca, and Mn compared to turnip greens (Table 5).

Light treatments significantly influenced the concentration of Mg, Ca, Mn, Fe, Ni, and P. In particular, R light increased the concentrations of Mg, Mn, Fe, and Ni, while W light increased the concentrations of Ca and P (Table 5).

The effect of LED treatment was more pronounced for Mg and the microelements (Mn, Fe, and Cu), which were significantly higher in turnip greens under the red LED. Amaranth grown under the red and blue LED showed a high Fe concentration. No significant differences were observed for Ni, Zn, and P (Table 5). 

### 2.7. RGB Color Analysis

The color analysis of microgreens showed that light significantly affected the RB components of amaranth, while no significant differences were observed for the RGB components of turnip greens (Figure 8). In amaranth the white light led to the highest R and B values, while the red light lowered the B value. In the comparison between species, the G component was significantly higher in turnip greens compared to amaranth (Figure 8).

When all the effects were summarized in a PCA score plot, differential reactions of amaranth and turnip greens to different light spectra were observed (Figure 9a,b). The first two PCs were related with eigen values >1 and explained more than 90% of the total variance, with PC1 and PC2 accounting for 56.4% and 43.6% for amaranth, and 54.1% and 45.9% for turnip greens. In amaranth, we identified four groups of positively correlated variables: (1) the group in the upper left quadrant, which included Chl *b*, sugars, and Na; (2) the group in the upper right quadrant, which included carotenoids, total Chl, Chl *a*, Asc, DPPH, TPC, Ca, and Cu; (3) the group clustered in the lower right quadrant, which included nitrates and most mineral elements (Fe, Mg, Zn, Mn, and Ni); (4) the group in the lower left quadrant, which included P, K, FW, and % DW (Figure 9a). 

For turnip greens, we identified the following groups: (1) the group in the upper left quadrant, which included K and Na; (2) the group in the upper right quadrant, which included sugars, nitrates, % DW, and some mineral elements (Ca, Zn, and Mn); (3) the group clustered in the lower right quadrant, which included carotenoids, Chl *a*, H/DW, FW, and most mineral elements (Cu, Ni, Fe, Mg, and Na); (4) the group in the lower left quadrant, which included antioxidant activity (DPPH and TPC), Chl *b*, total Chl, and Asc (Figure 9b).

Plants of amaranth grown under red LED, positioned in the lower left quadrant of the PCA score plot, exhibited a higher concentration of P and K, whilst those grown under blue LED, positioned in the upper right quadrant, were characterized by higher total Chl, Chl *a*, and antioxidant activity. Plants of turnip greens, grown under blue LED, positioned in lower left quadrant, were characterized by higher antioxidant level (TPC and Asc), antioxidant activity (DPPH), and total Chl, whilst those grown under red LED, positioned in the lower right quadrant of the PCA score plot, showed a higher content of mineral elements (Cu, Ni, Fe, Mg, and Na).

The PCA analysis reported in the present study could, therefore, help to better understand the influence of LED treatments on morphological and nutraceutical characteristics of the two studied species.

## 3. Discussion 

The results from this study revealed that the growth of hypocotyls in microgreens was affected by the quality of light. It has been reported in the literature how the hypocotyl growth may be influenced by artificial lights [33]; this aspect is relevant because the hypocotyls represent one of the main edible parts of sprouts and microgreens. To facilitate the machine harvest for labor savings, the height of microgreens needs to reach ~5 cm. The two studied microgreens exhibited a very different hypocotyl height. Among the three LED lights tested, blue and, to a lesser extent, red light seemed to be more effective than white light in promoting fresh biomass accumulation and hypocotyl growth. Similar significant increases in hypocotyl and shoot dry and fresh weight under monocromatic blue and red light were reported in microgreens of mustard and kale [34]. The blue LED, compared with the combined red and blue LED, was reported to increase the hypocotyl length of buckwheat sprouts [35]. Similarly, compared with the white LED, both blue and red LEDs were able to significantly increase the stem length of pea microgreens [36].

The leaf color of the two species, red (in amaranth) and green (in turnip greens), probably modifies the response to light spectra, as also observed in two cultivars of lettuce differing in leaf color (red and green) [37]. The PCA scatterplot clearly evidenced the differences between amaranth and turnip greens cultivated under different LED spectra (Appendix A).

Unlike fresh biomass, no significant difference between blue and white light was detected for dry biomass, revealing a higher water content in the plantlets grown under blue light. Our results partially differed from those obtained with other two leafy species, lettuce [38] and rocket [12], where better results in terms of fresh and dry biomass were obtained under red light, along with no difference in plant biomass between blue and white light. Microgreens are plants with a short growth period; therefore, the light spectrum influences more photomorphogenesis than photosynthesis. Photomorphogenic processes activated by the blue photomorphogenetic (cryptochromes, phytochromes) receptors constitute a default developmental process triggered by blue light in sprouts and microgreens during their development from seeds to edible vegetable products. The effects of blue light in microgreens may be different from those of mature plants [33,39].

In this study, a significant rise in Chl *a* and total Chl content under blue light was observed, although only in amaranth. Blue LEDs, used alone or in combination with red light, were reported to increase the chlorophyll ratio [40,41] and the chlorophyll content [42] in different leafy species and microgreens [15].

Sugar content significantly changed only with the species, being more than 40% greater in turnip greens than in amaranth.

Nitrate concentration in fresh vegetables is an important qualitative feature since its intake at high levels is associated with increased probability for carcinogenic nitrosamine formation in the stomach [32,43]. Approximately 80% of human dietary nitrates comes from vegetables; therefore, a low nitrate accumulation in vegetables is a primary concern [33]. In this study, nitrates were much greater in amaranth than in turnip greens, but lower than the maximum levels in European Commission (EC) Regulation No. 1258/2011 [44]. Moreover, the response to light spectra was species-specific, with the nitrate content significantly enhanced by red light in amaranth and unaffected by light spectra in turnip greens. This study confirms that nitrate accumulation capacity is a trait strongly associated with the genetic background of plants [21], even among genera within the same family. Contrasting results were reported in the literature for nitrate accumulation in response to red light; according to our results, an enhancement induced by monochromatic red light was found in mustard (*Brassica juncea* ‘Red Lace’) by Brazaityté et al. [34]. Conversely, a reduced nitrate content under a red LED was reported in *Perilla frutescens* (L.) and radish microgreens.

The content in antioxidants is a very relevant quality index of sprouts and microgreens. The phenolic phytochemical accumulation can be stimulated by cultivation under different LEDs. In our study, blue light positively influenced total polyphenols, carotenoids, ascorbic acid (Asc), and antioxidant activity. As compared to white light, red light exerted similar or negative effects on the antioxidants with the only exception being Asc content. Previous studies showed that total phenolic content was significantly increased under a blue LED, as compared with white LEDs in Chinese kale and common buckwheat sprouts [45,46]. In Chinese kale sprouts, the highest antioxidant capacity was measured under a blue LED [33,46].

Increasing blue light dosage has been recognized to increase the level of phenolics in lamb lettuce [47], of phenols and antioxidant activity in pea sprouts [48], and of carotenoids in some microgreens [15]. Blue light is reported to stimulate the accumulation of carotenoids via cryptochromes [49,50]. Carotenoids play a relevant role in photoprotective efficiency in plants [51,52]. Carotenoids protect plants from photo-oxidative damage through thermal dissipation by means of the xanthophyll cycle (converting violaxanthin to zeaxanthin) [53]. β-Carotene directly participates in light absorption, absorbing light in the blue region at 448 and 454 nm [49].

Differences between red and green lettuce [28,54] and basil [55] in growth, antioxidant levels, and photosynthetic response to red LED parameters were reported, which highlighted that red (purple) cultivars are less sensitive to environmental impacts. Similarly, in our study, according the PCA analysis, the two species with red (amaranth) and green (turnip greens) leaves showed a distinct response under the same lighting conditions.

## 4. Materials and Methods

This study was conducted on two microgreens: amaranth (*Amaranthus tricolor* L.) and turnip greens (*Brassica rapa* L. subsp. *oleifera* (DC.) Metzg) (CN Seeds, Ltd., Pymoor, Ely, Cambridgeshire, UK).

The experiment was performed in a controlled-environment growth chamber. Day and night temperature was maintained at 24 ± 2 °C within a 16/8 h light/dark photoperiod and a relative humidity of 50/60% was maintained. During the experiments, the air temperature and relative humidity (RH%) were measured using a meteo station (Avidsen Italia). Plantlets were grown in sowing substrate (‘Brill^®^ Semina Bio’, Agrochimica S.p.A., Bolzano, Italy) and vermiculite in containers (14 × 9 cm) for 10 days from sowing to harvest.

Three containers (i.e., three replicates) were used for each experimental treatment. 

Light-emitting diode (LED)-based lighting units, consisting of commercially avail-able LEDs with emission wavelengths of (1) white LED (W) (LEDW—blue 21%; green 38%; red 35%; dark red 6%—Grow Light C65 NS12—Valoya Oy Helsinki, Finland), (2) blue LED (B) (LEDR/B, BS Biosystem, Catania), and (3) red LED (R) (100% BS Biosystem, Catania), were used for microgreen lighting. The measured photosynthetic photon flux density (PPFD) sources (i.e., at the pot top level) were 200 ± 5 µmol for all the sectors. Spectral outputs from the various LED lamps were verified using a calibrated spectroradiometer LI-190R (LiCor, Inc., Lincoln, NE, USA, LICOR Biosciences).

### 4.1. Chemicals and Reagents 

Analytical reagent-grade chemicals and bi-distilled water were used throughout this experiment. Methanol used was of HPLC-grade, ≥99.9%, CHROMASOLV™ (Honeywell Riedel-de Haën™); KNO_3_, acetone (Multisolvent^®^ HPLC grade), NaOH, and H_2_SO_4_ were purchased from Merck KGaA, Darmstadt, Germany. Methyl viologen, oxalic acid anhydrous 99%, salicylic acid acs 99%, sodium carbonate (Na_2_CO_3_), glucose solution, anthrone 97%, l-ascorbic acid, Folin–Ciocâlteu reagent, DPPH^•^ radical reagent, Trolox, and gallic acid were purchased from Merck KGaA, Darmstadt, Germany. Standard solutions were prepared with bi-distilled water.

### 4.2. Measurement and Data Collection of Growth Parameters 

At harvest time, morphological parameters, seedling fresh biomass (g), seedling dry biomass (g), and seedling height (cm) were measured. The height (H), fresh weight (FW), and dry weight (DW) were determined on 15 seedlings, randomly selected within each container. The weight was expressed as micrograms per seedling. The dry biomass (DW) of the plants was obtained by putting weighed samples in a thermo-ventilated oven at 70 °C until they reached a constant weight. Stem and leaves were immersed in liquid nitrogen and kept at −80 °C for phytochemical analysis. The plant height/plant dry weight ratio (H/DW, cm·mg^−1^) was also calculated. For all chemical analysis, three replicates were performed. 

### 4.3. Chlorophyll and Carotenoid Pigments

The contents of chlorophyll (Chl *a*, Chl *b*, and total Chl) and carotenoids was analyzed using the spectrophotometric method. Samples of 150 mg were extracted using 99% methanol and incubated in dark room (4 °C for 24 h). The absorbance of samples was read at 665.2 nm, 652.4 nm, and 470 nm, respectively, for Chl *a*, Chl *b*, and carotenoids in a spectrophotometer (7315 Spectrophotometer, Jenway, Staffordshire, UK). Chlorophyll and carotenoid contents were calculated as described by Lichtenthaler et al. [56].

Chl *a* = 16.75*A*_665.2_ − 9.16*A*_652.4_.

Chl *b* = 34.09*A*_652.4_ − 15.28*A*_665.2_.

Carotenoids = (1000*A*_470_ − 1.63Chl *a* − 104.96Chl *b*)/221.

### 4.4. Total Sugars

The total sugars were determined spectrophotometrically following the anthrone method with slight modifications [57]. The anthrone reagent (10.3 mM) was prepared by dissolving anthrone in ice-cold 95% H_2_SO_4_. The reagent was left to stand for 30–40 min before use. Then, 1 g of fresh sample was extracted in 3 mL of distilled water and centrifuged at 3000× *g* for 15 min at room temperature (RT). Next, 0.5 mL of extract was placed on top of 2.5 mL of anthrone reagent incubated in ice for 5 min. The reactions were heated to 95 °C for 10 min and left to cool in ice. The absorbance was read at 620 nm. A calibration curve was generated using a glucose solution (0 to 0.05 mg·mL^−1^) (*R*^2^ = 0.9995).

### 4.5. Nitrate Concentrations

Nitrate concentrations were determined following the salicyl sulfuric acid method [58]. First, 1 g of fresh sample was homogenized in 3 mL of distilled water and then centrifuged (4000 rpm, 15 min), collecting the supernatant. Then, 20 µL of extract was added to 80 μL of 5% salicylic acid in sulfuric acid and to 3 mL of NaOH 1.5 N. The samples were cooled, and the spectrophotometer readings were read at 410 nm. A calibration curve was generated using a KNO_3_ standard (0, 1, 2.5, 5, 7.5, and 10 mM KNO_3_) (*R*^2^ = 0.9918).

### 4.6. Ascorbic Acid Analysis

The ascorbic acid content was determined using a spectrophotometric method [59]. Fresh plant tissue (1 g) was homogenized in 10 mL of 5% oxalic acid and then centrifuged (5 min, 4000 rpm). The extract (1 mL) was added to 2 mL of 0.1% methyl viologen and 2 mL of 2 mol·L^−1^ NaOH. The colored radical ion was read at 600 nm against the radical blank. The concentration of ascorbic acid was calculated as a function of the values obtained from the l-ascorbic acid standard curve (100–500 µg·mL^−1^) (*R*^2^ = 0.9907). Results were expressed as mg·g^−1^ fresh weight.

### 4.7. Total Phenolic Compounds and 2,2-Diphenyl-1-picrylhydrazyl (DPPH) Radical-Scavenging Activity

First, 1 g FW of sample was homogenized in a solution containing 50% acetone and 50% water (1:10). The samples were vortexed and incubated for 15 h at 20 °C. Then, 100 μL of supernatant was mixed with 0.5 mL of Folin–Ciocâlteu reagent (Sigma-Aldrich, Italy) and 6 mL of distilled water. Next, 1.5 mL of Na_2_CO_3_ (20%) was added, before incubating at 20 °C for 2 h. The absorbance was read at 765 nm. The concentration of total phenolic compounds was calculated as a function of the values obtained from the gallic acid standard curve (0, 50, 100, 250, and 500 mg·L^−1^) (*R*^2^ = 0.9954). The total phenolic content was expressed as mg·100 g^−1^ gallic acid equivalent.

The antioxidant activity was determined using DPPH. About 1 g of fresh weight was mixed with 1.5 mL of methanol solution (80%), sonicated for 30 min, and centrifuged (10 min, 5 °C, 5000× *g*). Then, 0.01 mL of supernatant was mixed with 1.4 mL of 150 μM DPPH solution in methanol and water (95:5), before incubating for 30 min in the dark. The sample was read at 517 nm. The antioxidant activity was calculated as a function of the values obtained from the Trolox standard curve (0 to 0.5 mg·mL^−1^) (*R*^2^ = 0.9995). DPPH scavenging activity values were expressed as Trolox equivalent antioxidant activity (mg TE·100 g^−1^).

### 4.8. Meso and Micro Elements

Meso and micro element (Na, Mg, K, Ca, Mn, Fe, Ni, Cu, Zn, and P) concentrations were determined on oven-dried samples (80 °C for 48 h). Samples of 300 mg of dry matter were mineralized at 120 °C in 5 mL of 14.4 M HNO_3_, clarified with 1.5 mL of 33% H_2_O_2_, and dried at 80 °C. The mineralized material was solubilized in 5 mL of 1 M HNO_3_ and filtered on a 0.45 μm nylon membrane. Mineral elements were measured using inductively coupled plasma mass spectroscopy (ICP-MS; Varian 820-MS, ICP Mass Spectrometer). Concentrations of mineral elements were expressed on a dry weight basis. 

### 4.9. RGB Color Analysis

Photos of microgreens grown in different treatments were taken at 30 cm distance. The colors of the photos were analyzed using online tools (https://imagecolorpicker.com/, accessed on 8 June 2021) for the measurements the RGB components. 

### 4.10. Statistical Analysis

The experiment was performed using a completely randomized design. Three biological replicates were used for the analysis. Data were subjected to two-way ANOVA, and differences among means were determined using Tukey’s post hoc test (*p* ≤ 0.05) All statistical analyses were performed using CoStat release 6.311 (CoHort Software, Monterey, CA, USA). The principal component loading plot and scores of PCA were obtained using Minitab 16, LLC. The data presented in the figures are the means ± *se* (Graphpad 7.0). 

## 5. Conclusions

The results obtained with our trial indicate that the following:-blue light was particularly effective in enhancing the growth and nutritional characteristics (particularly antioxidant activity) of the two studied microgreens as compared to the more traditionally used white light;-red light seemed to be more effective than white light in promoting fresh biomass accumulation and hypocotyl growth. However, its effects on nutraceutical characteristics were quite different for the two genotypes, since it did not influence those of turnip greens but worsened those of amaranth (see nitrates, nickel, and total polyphenol contents) as compared to the other lights;-the response to the spectral system is typically species-specific; for this reason, it is possible to adopt a specific light formula that allows maximizing both plant growth and nutritional quality, thereby enhancing the microgreen industry.

## Figures and Tables

**Figure 1 plants-10-01584-f001:**
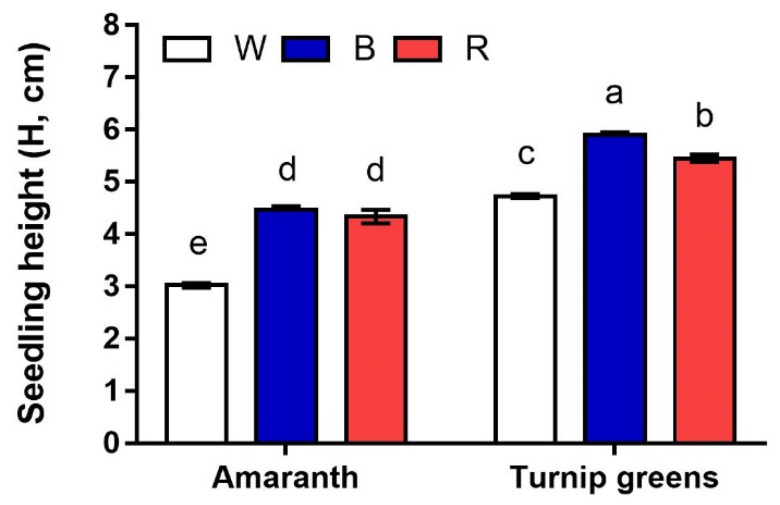
Interaction effect of *species* × *light treatment* (W = white, B = blue, R = red) on seedling height (H, cm) of microgreens. Data are means ± standard error (*n* = 3). Three biological replicates were used for the measurements. Different letters indicate significance at *p* ≤ 0.05 according to Tukey’s test.

**Figure 2 plants-10-01584-f002:**
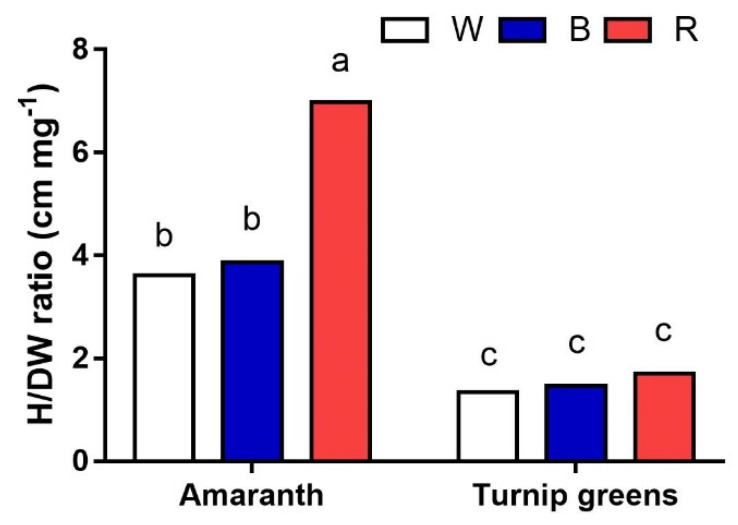
Interaction effect of *species* × *light treatment* (W = white, B = blue, R = red) on plant height/dry biomass ratio (H/DW, cm·mg^−1^) of microgreens. Data are means ± standard error (*n* = 3). Three biological replicates were used for the measurements. Different letters indicate significance at *p* ≤ 0.05 according to Tukey’s test.

**Figure 3 plants-10-01584-f003:**
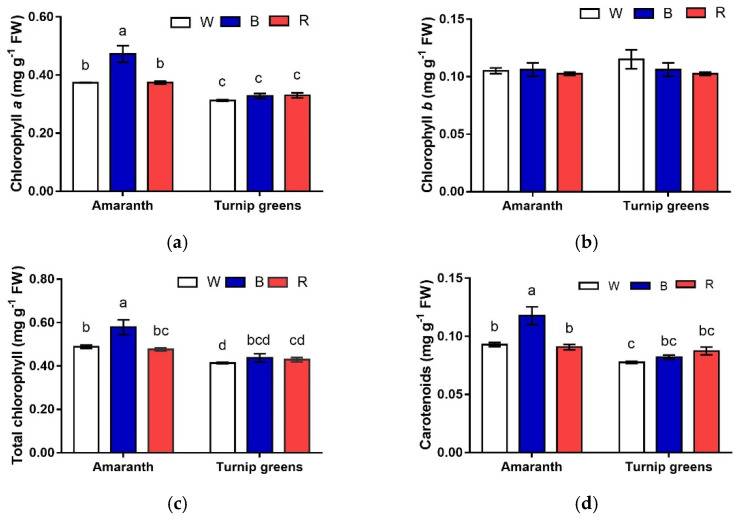
Interaction effect of *species* × *light treatment* (W = white, B = blue, R = red) on chlorophyll (*a* (**a**), *b* (**b**), and total (**c**)) and carotenoids (**d**) of microgreens. Data are means ± standard error (*n* = 3). Different letters indicate significance at *p* ≤ 0.05 according to Tukey’s test.

**Figure 4 plants-10-01584-f004:**
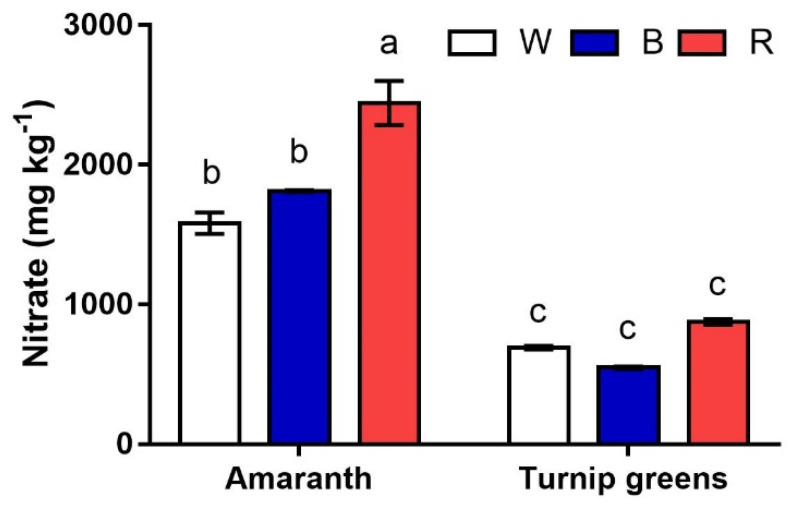
Interaction effect of *species* × *light treatment* (W = white, B = blue, R = red) on the nitrate content (ppm) of microgreens. Data are means ± standard error (*n* = 3). Three biological replicates were used for the analysis. Different letters indicate significance at *p* ≤ 0.05 according to Tukey’s test.

**Figure 5 plants-10-01584-f005:**
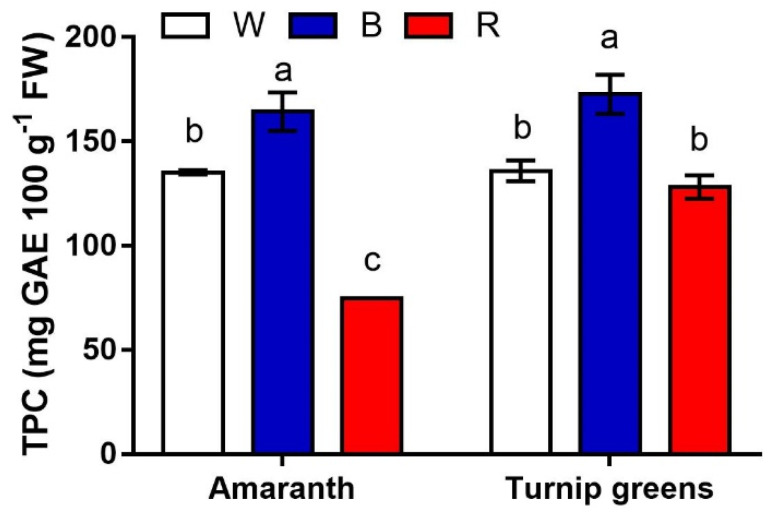
Interaction effect of *species* × *light treatment* (W = white, B = blue, R = red) on TPC (mg GAE·100 g^−1^ FW) of microgreens. Data are means ± standard error (*n* = 3). Three biological replicates were used for the analysis. Different letters indicate significance at *p* ≤ 0.05 according to Tukey’s test.

**Figure 6 plants-10-01584-f006:**
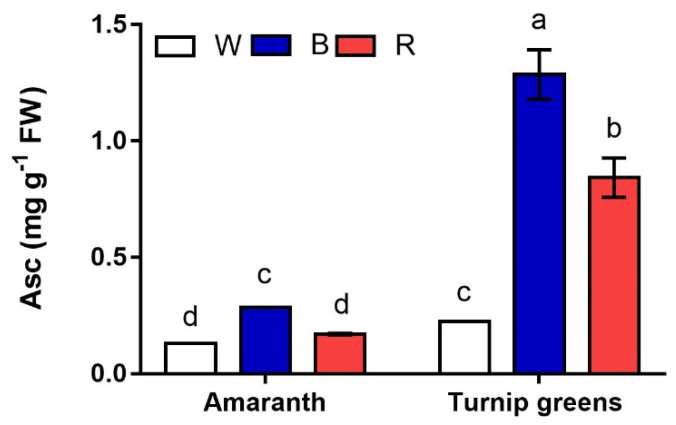
Interaction effect of *species* × *light treatment* (W = white, B = blue, R = red) on ascorbic acid (Asc, mg·100 g^−1^ FW) of microgreens. Data are means ± standard error (*n* = 3). Three biological replicates were used for the analysis. Different letters indicate significance at *p* ≤ 0.05 according to Tukey’s test.

**Figure 7 plants-10-01584-f007:**
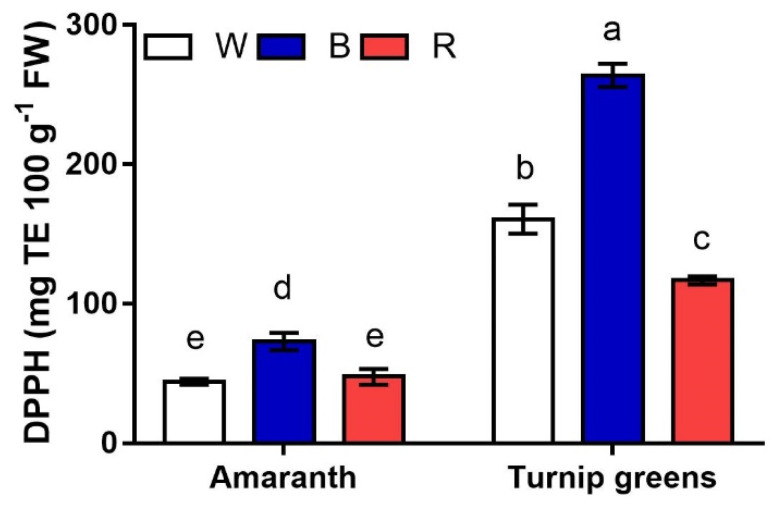
Interaction effect of *species* × *light treatment* (W = white, B = blue, R = red) on the antioxidant activity (DPPH, mg TE·100 g^−1^ FW) of microgreens. Data are means ± standard error (*n* = 3). Three biological replicates were used for the analysis. Different letters indicate significance at *p* ≤ 0.05 according to Tukey’s test.

**Figure 8 plants-10-01584-f008:**
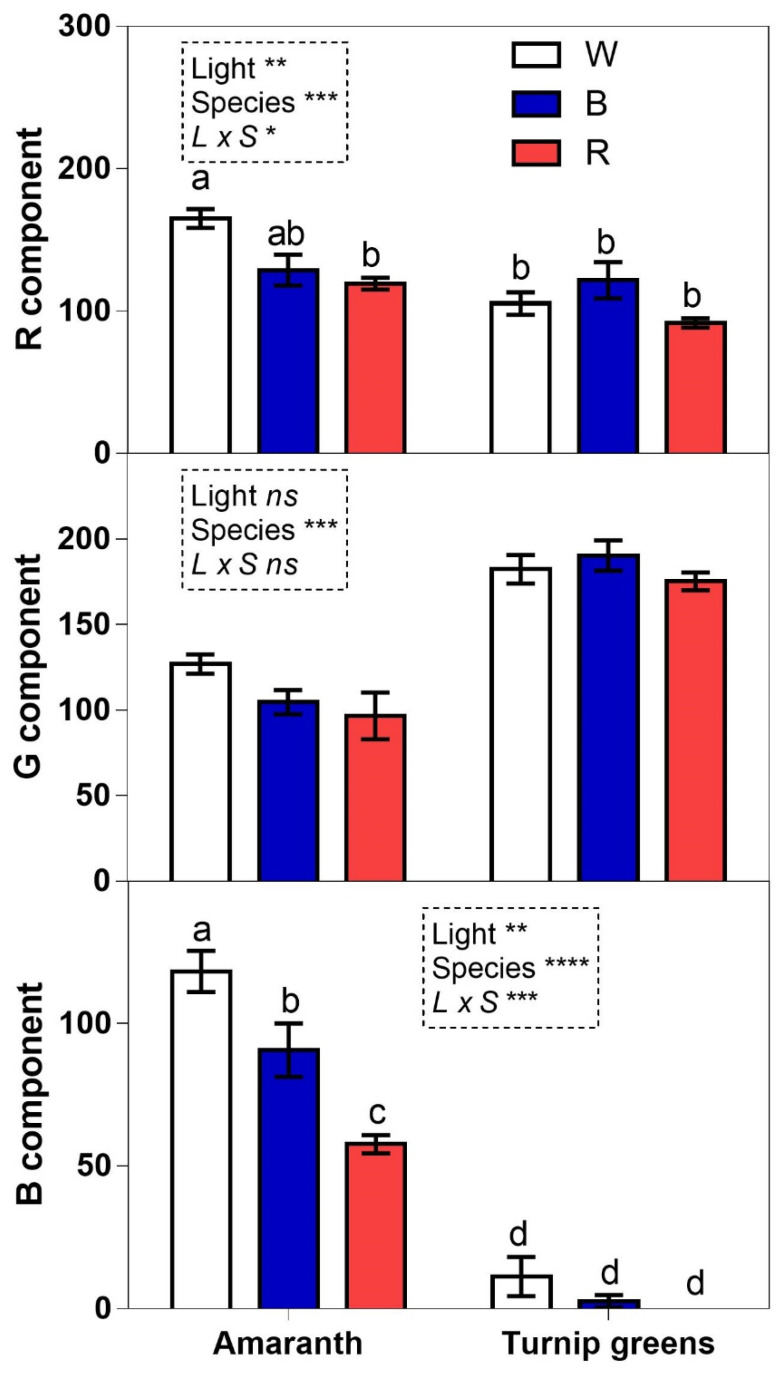
RGB component analysis of microgreen photos grown under different light conditions: white (W), blue (B), and red (R). Values are means with standard errors (*n* = 4). Four biological replicates were used for the analysis. Data were subjected to two-way ANOVA. Differences among means were determined using Tukey’s test. Different letters highlight significant differences at *p* ≤ 0.05; ns not significant, significant at *p* ≤ 0.05 (*), 0.01 (**), and 0.001 (***).

**Figure 9 plants-10-01584-f009:**
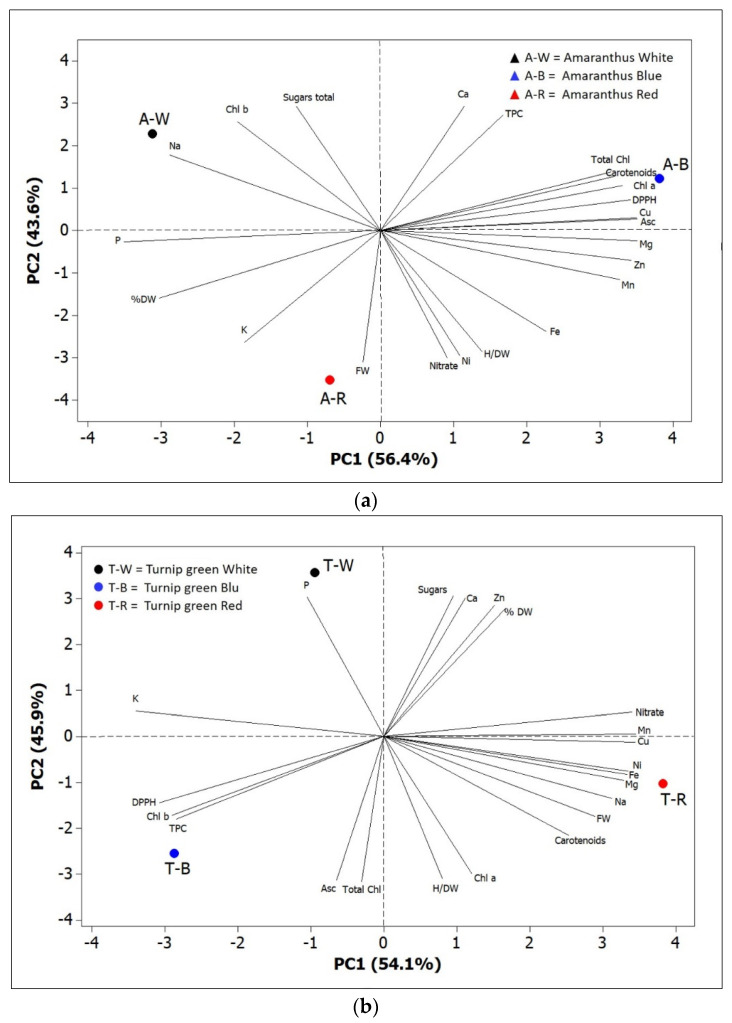
Principal component loading plot and scores of PCA fresh weight and dry biomass, H/DW, photosynthetic pigments (Chl *a*, Chl *b*, total Chl, and carotenoids), mineral concentrations (nitrate, Na, Mg, P, K, Ca, Mn, Fe, Ni, Cu, and Zn), DPPH, TPC, total sugars, Asc, and total phenolic concentrations for amaranth (**a**) and turnip greens (**b**) as modulated by LED treatments. W = white LED treatment; B = blue LED treatment; R = red LED treatment.

**Table 1 plants-10-01584-t001:** Main effects of species (amaranth and turnip greens) and LED treatment (W = white, B = blue, R = red) on plant height, fresh biomass, and dry biomass percentage of microgreens.

		Seedling Height (H, cm)	Fresh Biomass (FW, mg·Plant^−1^)	Dry Biomass (DW, %)
Species (*S*)	Amaranth	3.9 ± 0.2 ^b^	18.5 ± 1.8 ^b^	5.4 ± 0.4
	Turnip greens	5.4 ± 0.2 ^a^	66.4 ± 2.8 ^a^	5.2 ± 0.3
LED treatments (*L*)	W	3.9 ± 0.4 ^c^	37.1 ± 9.9 ^b^	5.9 ± 0.3 ^a^
	B	5.2 ± 0.4 ^a^	50.1 ± 11.8 ^a^	5.9 ± 0.3 ^a^
	R	4.9 ± 0.2 ^b^	40.2 ± 10.8 ^b^	4.2 ± 0.1 ^b^
*Significance*	*S*	***	***	ns
	*L*	***	***	***
	*S* × *L*	**	ns	ns

Values (mean ± *se*) within each column, followed by the same letter, do not significantly differ at *p* ≤ 0.05 according to Tukey’s test; ns = not significant; significant at *p* ≤ 0.01 (**) and 0.001 (***). Three biological replicates were used for measurements (*n* = 3).

**Table 2 plants-10-01584-t002:** Main effects of species (amaranth and turnip greens) and LED treatment (W = white, B = blue, R = red) on chlorophyll *a*, *b*, chlorophyll *a*/*b* ratio (Chl *a*/Chl *b*), total chlorophyll, carotenoids, and chlorophyll/carotenoid ratio (Chl/Car) of microgreens.

		Chl *a* (mg·g^−1^ FW)	Chl *b* (mg·g^−1^ FW)	Chl *a*/Chl *b* (mg·g^−1^ FW)	Total Chl (mg·g^−1^ FW)	Carotenoids (mg·g^−1^ FW)	Chl/Car (mg·g^−1^ FW)
Species (*S*)	Amaranth	0.41 ± 0.0 ^a^	0.11 ± 0.0 ^a^	3.79 ± 0.18 ^a^	0.51 ± 0.0 ^a^	0.10 ± 0.00 ^a^	5.1 ± 0.1
	Turnip greens	0.32 ± 0.0 ^b^	0.10 ± 0.0 ^b^	3.20 ± 0.20 ^b^	0.43 ± 0.0 ^b^	0.08 ± 0.00 ^b^	5.2 ± 0.1
LED treatment (*L*)	W	0.34 ± 0.1	0.11 ± 0.0	3.18 ± 0.11 ^b^	0.45 ± 0.0 ^b^	0.08 ± 0.00 ^b^	5.3 ± 0.1
	B	0.40 ± 0.0	0.11 ± 0.0	3.74 ± 0.33 ^a^	0.51 ± 0.0 ^a^	0.10 ± 0.01 ^a^	5.1 ± 0.2
	R	0.35 ± 0.0	0.10 ± 0.0	3.49 ± 0.08 ^ab^	0.45 ± 0.0 ^b^	0.09 ± 0.00 ^b^	5.1 ± 0.1
*Significance*	*S*	***	***	***	***	***	ns
	*L*	ns	ns	**	**	**	ns
	*S* × *L*	***	ns	ns	*	**	ns

Values (mean ± *se*) within each column, followed by the same letter, do not significantly differ at *p* ≤ 0.05 according to Tukey’s test; ns = not significant; significant at *p* ≤ 0.05 (*), 0.01 (**), and 0.001 (***). Three biological replicates were used for the analysis (*n* = 3).

**Table 3 plants-10-01584-t003:** Main effects of species (amaranth and turnip greens) and LED treatment (W = white, B = blue, R = red) on total sugars and nitrate content of microgreens.

		Total Sugars (mg·g^−1^ FW)	Nitrate (mg·kg^−1^)
Species (*S*)	Amaranth	0.7 ± 0.0 ^b^	1990.9 ± 140.3 ^a^
	Turnip greens	1.3 ± 0.0 ^a^	704.9 ± 48.0 ^b^
LED treatment (*L*)	W	1.0 ± 0.4	1137.1 ± 202.5 ^b^
	B	1.0 ± 0.4	1247.1 ± 318.7 ^b^
	R	0.9 ± 0.4	1659.5 ± 357.7 ^a^
*Significance*	*S*	***	***
	*L*	ns	***
	*S* × *L*	Ns	***

Values (mean ± *se*) within each column, followed by the same letter, do not significantly differ at *p* ≤ 0.05 according to Tukey’s test; ns = not significant; significant at *p* ≤ 0.001 (***). Three biological replicates were used for the analysis (*n* = 3).

**Table 4 plants-10-01584-t004:** Main effects of species (amaranth and turnip greens) and LED treatment (W = white, B = blue, R = red) on total phenolic content (TPC), ascorbic acid (Asc), and antioxidant activity (DPPH) of microgreens.

		TPC (mg GAE·100 g^−1^ FW)	Asc (mg·g^−1^ FW)	DPPH (mg TE·100 g^−1^ FW)
Species (*S*)	Amaranth	124.8 ± 13.5 ^b^	0.20 ± 0.0 ^b^	54.6 ± 5.2 ^b^
	Turnip greens	145.6 ± 7.7 ^a^	0.78 ± 0.2 ^a^	180.5 ± 22.2 ^a^
LED treatment (*L*)	W	135.6 ± 2.3 ^b^	0.79 ± 0.2 ^a^	102.4 ± 26.5 ^b^
	B	168.6 ± 6.2 ^a^	0.20 ± 0.0 ^c^	168.5 ± 42.9 ^a^
	R	104.4 ± 12.2 ^c^	0.51 ± 0.1 ^b^	82.3 ± 15.7 ^c^
*Significance*	*S*	**	***	***
	*L*	***	***	***
	*S* × *L*	**	***	**

Values (mean ± *se*) within each column, followed by the same letter, do not significantly differ at *p* ≤ 0.05 according to Tukey’s test; significant at *p* ≤ 0.01 (**) and 0.001 (***). Three biological replicates were used for the analysis (*n* = 3).

**Table 5 plants-10-01584-t005:** Results of the multifactorial ANOVA for sodium (Na), magnesium (Mg), potassium (K), calcium (Ca), manganese (Mn), iron (Fe), nickel (Ni), copper (Cu), zinc (Zn), and phosphorus (P) concentrations of microgreens.

			Na (g·kg^−1^ DW)	Mg (g·kg^−1^ DW)	K (g·kg^−1^ DW)	Ca (g·kg^−1^ DW)	Mn (mg·kg^−1^ DW)	Fe (mg·kg^−1^ DW)	Ni (mg·kg^−1^ DW)	Cu (mg·kg^−1^ DW)	Zn (mg·kg^−1^ DW)	P (g·kg^−1^ DW)
Species (*S*)		Amaranth	2.5 ± 0.1 ^b^	11.2 ± 0.3 ^a^	94.2 ± 1.7 ^a^	8.1 ± 0.2 ^b^	48.8 ± 1.4 ^b^	1.7 ± 0.2	9.5 ± 1.6	27.1 ± 1.3 ^a^	102.7 ± 1.5 ^a^	12.2 ± 0.2 ^a^
		Turnip greens	4.8 ± 0.1 ^a^	9.5 ± 1.1 ^b^	72.0 ± 1.4 ^b^	10.9 ± 0.2 ^a^	54.9 ± 4.5 ^a^	2.1 ± 0.5	8.2 ± 1.4	23.7 ± 1.7 ^b^	77.5 ± 2.0 ^b^	9.3 ± 0.1 ^b^
			*p* ≤ 0.001	*p* ≤ 0.05	*p* ≤ 0.001	*p* ≤ 0.001	*p* ≤ 0.01	*p* ≥ 0.05	*p* ≥ 0.05	*p* ≤ 0.05	*p* ≤ 0.001	*p* ≤ 0.001
LED treatment (*L*)		W	3.6 ± 0.4	8.9 ± 0.6 ^b^	83.9 ± 4.4	9.9 ± 0.7 ^a^	47.5 ± 1.7 ^b^	1.1 ± 0.0 ^b^	5.2 ± 0.8 ^b^	23.4 ± 1.4	91.7 ± 4.9	11.0 ± 0.7 ^a^
		B	3.5 ± 0.5	9.9 ± 1.1 ^b^	83.0 ± 3.7	9.4 ± 0.4 ^ab^	47.6 ± 2.5 ^b^	1.5 ± 0.2 ^b^	7.3 ± 1.2 ^b^	24.9 ± 2.5	88.0 ± 7.2	10.4 ± 0.5 ^b^
		R	3.7 ± 0.6	12.2 ± 0.8 ^a^	82.6 ± 7.2	9.2 ± 0.8 ^b^	60.4 ± 5.3 ^a^	3.0 ± 0.5 ^a^	14.1 ± 0.8 ^a^	27.8 ± 1.5	90.6 ± 5.7	10.8 ± 0.7 ^ab^
			*p* ≥ 0.05	*p* ≤ 0.01	*p* ≥ 0.05	*p* ≤ 0.05	*p* ≤ 0.001	*p* ≤ 0.001	*p* ≤ 0.001	*p* ≥ 0.05	*p* ≥ 0.05	*p* ≤ 0.05
S × L	Amaranth	W	2.7 ± 0.0 ^b^	10.2 ± 0.1 ^ab^	93.6 ± 1.6 ^a^	8.3 ± 0.3 ^c^	43.8 ± 0.5 ^b^	1.1 ± 0.1 ^c^	4.6 ± 2.1	24.8 ± 4.2 ^ab^	101.7 ± 6.5	12.6 ± 0.5
		B	2.4 ± 0.1 ^b^	12.3 ± 0.3 ^a^	90.8 ± 3.8 ^a^	8.5 ± 0.4 ^c^	52.7 ± 0.7 ^b^	1.9 ± 0.1 ^b^	9.2 ± 2.2	30.2 ± 0.2 ^a^	103.7 ± 5.6	11.6 ± 0.3
		R	2.4 ± 0.3 ^b^	11.1 ± 0.5 ^a^	98.2 ± 6.7 ^a^	7.5 ± 0.8 ^c^	49.9 ± 2.9 ^b^	2.1 ± 0.3 ^b^	14.5 ± 2.7	26.2 ± 4.2 ^ab^	102.8 ± 0.6	12.3 ± 0.5
	Turnip greens	W	4.6 ± 0.0 ^a^	7.5 ± 0.2 ^b^	74.1 ± 0.1 ^b^	11.5 ± 0.5 ^a^	51.2 ± 2.0 ^b^	1.2 ± 0.1 ^c^	5.7 ± 1.8	22.0 ± 2.6 ^ab^	81.7 ± 4.4	9.4 ± 0.2
		B	4.7 ± 0.2 ^a^	7.7 ± 1.2 ^b^	75.1 ± 1.1 ^b^	10.2 ± 0.4 ^b^	42.6 ± 3.9 ^b^	1.2 ± 0.1 ^c^	5.3 ± 2.0	19.6 ± 3.4 ^b^	72.3 ± 3.1	9.3 ± 0.2
		R	5.0 ± 0.1 ^a^	13.3 ± 2.6 ^a^	66.9 ± 2.3 ^b^	10.9 ± 0.3 ^ab^	70.9 ± 8.9 ^a^	3.9 ± 1.1 ^a^	13.6 ± 1.2	29.4 ± 3.0 ^a^	78.4 ± 6.3	9.3 ± 0.3
			*p* ≤ 0.05	*p* ≤ 0.01	*p* ≤ 0.01	*p* ≤ 0.05	*p* ≤ 0.001	*p* ≤ 0.01	*p* ≥ 0.05	*p* ≤ 0.05	*p* ≥ 0.05	*p* ≥ 0.05

The mean values associated with the two factors and their interaction were evaluated according to Tukey’s test. Means significantly different are indicated by different letters; ns not significant, significant at *p* ≤ 0.05 (*), 0.01 (**), and 0.001 (***). Three biological replicates were used for the analysis (*n* = 3).

## Data Availability

Main data are contained within the article; further data presented in this study are available on request from the corresponding author.

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
