# Peer review of "Effects of Different Light Spectra on Final Biomass Production and Nutritional Quality of Two Microgreens"

_plants, 2021, doi:10.3390/plants10081584_

Round 1

Reviewer 1 Report

In the presented manuscript, the Authors attempted to evaluate the effect of the light spectrum on biomass and the composition of phytochemicals in two types of microgreens.

The goal and the hypothesis were clearly stated. The language used in the article is correct and easy to understand. The results are supported by discussion, but the conclusions need to be improved. The topic is interesting - in line with the current trends. I have put all my comments in the attached file.

In general, only the methodology raises my doubts. Most of the analytical methods used are spectrophotometric (e.g. determination of chlorophylls, sugars or TPC with Folin's reagent). By current standards, these methods are outdated and inaccurate. The work would gain significantly in quality by adding the HPLC / UPLC assay. 

Author Response

reviewer 1

Comments and Suggestions for Authors

In the presented manuscript, the Authors attempted to evaluate the effect of the light spectrum on biomass and the composition of phytochemicals in two types of microgreens.

The goal and the hypothesis were clearly stated. The language used in the article is correct and easy to understand. The results are supported by discussion, but the conclusions need to be improved. The topic is interesting - in line with the current trends. I have put all my comments in the attached file.

Dear reviewer,

The authors would like to for the positive comments and evaluation. The manuscript has been accordingly revised. Corrections and suggestions have been implemented in the current version of the manuscript. We hereby provide a point-by-point answer.

The authors

In general, only the methodology raises my doubts. Most of the analytical methods used are spectrophotometric (e.g. determination of chlorophylls, sugars or TPC with Folin's reagent). By current standards, these methods are outdated and inaccurate. The work would gain significantly in quality by adding the HPLC / UPLC assay.

A.A. The methods used are spectrophometric since the quality evaluation was carried out considering the amount of the compounds considered. We agree with the referee that the HPLC analyses are more accurate, however, for nutritional point of view we focused on total amounts.

Line 15: ±

A.A.: Sorry for the mistakes. The corrections was done

Line 27: chlorophylls

A.A.: Done

Table 1: if we provide units in the title of the table, please do not duplicate them in the table, and vice versa. superscript letters

A.A.: The units of measurement have been deleted from the title of the tables and all letters have been superscribed.

Table 2: the same as in Table 1 - I suggest writing units in the title, then it will be more legible. Superscript, and further in the text

Table 4: as above

Table 5: as above

A.A.: Changes have been done in all tables

Line 270: Please provide a detailed description of the charts: what do the colors of the dots mean? why the letters "R" appear next to some of them? In their present form, they are hardly legible and difficult to interpret. There is no description with the interpretation of the results.

A.A.: The description of the legend has been modified according your comments.

Line 346: Please provide the list of reagents used in the tests together with the name of the supplier.

A.A.: The list of reagents used has been added.

Line 376: Were calibration curves used in the calculations? If so, please provide their characteristics.

Line 381: Please provide formulas for calculations.

Line 384: Were calibration curves used in the calculations? If so, please provide their characteristics.

Line 388: Please provide the concentration range.

Line 390: Were calibration curves used in the calculations? If so, please provide their characteristics.

Line 395: Please provide the concentration range.

Line 397: the same comment as above

Line 401: Please provide an exact description; as above

A.A.: All the information has been added in the methodologies.

Line 419: Please provide the name of the device, manufacturer and city / country - use and standardize throughout the methodology. Please provide details of the analysis

A.A.: The details of the analysis have been added

Line 435: The conclusions are too general. Please refer to individual analyzes - I propose to list them in points

A.A.: The concluding sentence was modified according your suggestion.

Reviewer 2 Report

Dear Authors,

The article deals with an interesting and current problem of the influence of light on plant physiology. It includes a number of different measurements showing the functioning of young plants under the conditions of different spectral composition of light. It is written correctly and clearly. I have marked my small suggestions in the text. I also ask the authors to verify the article in terms of editing, as there are still some minor errors. I fully recommend publishing the article.

Sincerely Yours,

Reviewer

Author Response

Reviewer 2

The article deals with an interesting and current problem of the influence of light on plant physiology. It includes a number of different measurements showing the functioning of young plants under the conditions of different spectral composition of light. It is written correctly and clearly. I have marked my small suggestions in the text. I also ask the authors to verify the article in terms of editing, as there are still some minor errors. I fully recommend publishing the article.

Dear reviewer,

Thank you very much for the suggestions and recommendation. The manuscript has been accordingly revised.

Lines 75-77: In my opinion, this sentence in the introduction is not necessary.

A.A. The sentence was removed.

Line 246: ±se, in italic

A.A. Done

Line 449: Ward, J.M.; Cufr, C.A.; Denzel, M.A; Neff, M.M. this is correct citation. Please check THE AUTHOR GUIDELINES!

A.A. Done

Round 2

Reviewer 1 Report

I recommend the manuscript for publication as it stands. 

Author Response

Thanks for your favorable review